# An Empirical Study of the Accuracy-Robustness Trade-off and Training Efficiency in Robust Self-Supervised Learning

**Fatemeh Ghofrani**                                                     *ghofrani@email.sc.edu*
*Department of Computer Science and Engineering*
*University of South Carolina*

**Mehdi Yaghouti**                                                       *yaghouti@mailbox.sc.edu*
*Department of Computer Science and Engineering*
*University of South Carolina*

**Pooyan Jamshidi**                                                      *pjamshid@cse.sc.edu*
*Department of Computer Science and Engineering*
*University of South Carolina*

**Reviewed on OpenReview:** *https://openreview.net/forum?id=WTqHDiETg5*

## Abstract

Self-supervised learning (SSL) has made significant strides in learning image representations, yet its principles remain partially understood, particularly in adversarial scenarios. This work explores the interplay between SSL and adversarial training (AT), focusing on whether this integration can yield robust representations that balance computational efficiency, clean accuracy, and robustness. A major challenge lies in the inherently high cost of AT, which combines an inner maximization problem (generating adversarial examples) with an outer minimization problem (training representations). This challenge is exacerbated by the extensive training epochs required for SSL convergence, which become even more demanding in adversarial settings.

Recent advances in SSL, such as Extreme-Multi-Patch Self-Supervised Learning (EMP-SSL), have demonstrated that increasing the number of patches per image instance can significantly reduce the number of training epochs. Building on this, we introduce Robust-EMP-SSL, an extension of EMP-SSL specifically designed for adversarial training scenarios. Robust-EMP-SSL is a framework that leverages multiple crops per image to enhance data diversity, integrates invariance terms with regularization to prevent collapse, and optimizes adversarial training efficiency by reducing the required training epochs. By aligning these components, Robust-EMP-SSL enables the learning of robust representations while addressing the high computational costs and accuracy trade-offs inherent in adversarial training.

This study poses a central question: "How can multiple crops or diverse patches, combined with adversarial training strategies, achieve trade-offs between computational efficiency, clean accuracy, and robustness?"

Our empirical results show that Robust-EMP-SSL not only accelerates convergence, but also achieves a superior balance between clean accuracy and adversarial robustness, outperforming SimCLR, a widely used self-supervised baseline that, like other methods, relies on only two augmentations. Furthermore, we propose the Cost-Free Adversarial Multi-Crop Self-Supervised Learning (CF-AMC-SSL) method, which incorporates free adversarial training into the multi-crop SSL framework. CF-AMC-SSL demonstrates the potential to enhance both clean accuracy and adversarial robustness with reduced epoch conditions, further improving efficiency.

These findings highlight the potential of Robust-EMP-SSL and CF-AMC-SSL to make SSL more practical in adversarial scenarios, paving the way for future empirical explorations and real-world applications. Our code is released at `https://github.com/softsys4ai/CF-AMC-SSL`.

# 1 Introduction

In recent years, progress in self-supervised learning (SSL) (Balestriero et al., 2023) has produced representations that match or exceed those achieved by supervised learning in classification tasks (Chen et al., 2020; Grill et al., 2020). For instance, methods such as SimCLR (Chen et al., 2020) and BYOL (Grill et al., 2020) have demonstrated performance on par with supervised approaches. This advancement has led to state-of-the-art performance in various modern AI-enabled applications, including models such as BERT and GPT-3 (Brown et al., 2020; Devlin et al., 2018).

Despite these remarkable achievements, challenges remain to ensure the robustness and reliability of SSL methods, particularly when training is used to produce effective representations from unlabeled data. Among existing SSL approaches, **joint-embedding SSL methods** aim to produce consistent embeddings for different augmentations of the same image. However, a significant challenge in these methods is avoiding **collapse**, where the model produces identical representations regardless of the input, thus losing the ability to capture meaningful variations in the data.

To address this issue, several strategies have been proposed, which can be categorized into the following approaches: (i) **Contrastive Learning**: These methods, such as SimCLR (Chen et al., 2020), rely on distinguishing between similar and dissimilar samples to maintain meaningful and diverse representations. By incorporating negative samples, they ensure that the embeddings of different inputs are well separated in the representation space. (ii) **Non-Contrastive Methods**: Unlike contrastive approaches, methods such as Barlow Twins (Zbontar et al., 2021) and VICReg (Bardes et al., 2021) avoid the need for negative samples by introducing covariance regularization. This approach promotes diversity in embeddings while maintaining consistency for different augmentations, effectively preventing collapse.

Building on these strategies, **multi-crop techniques**, as introduced in SwAV (Caron et al., 2020), have emerged as a promising approach to further enhance SSL performance. SwAV leverages multiple views of an image at varying resolutions, clustering embeddings to align augmentations while avoiding trivial solutions. This innovative strategy not only prevents collapse but also improves the quality of learned representations by encouraging diversity and consistency.

While these advancements have significantly improved the quality and diversity of SSL representations, another pressing concern is the vulnerability of SSL methods to adversarial attacks (Ghofrani et al., 2023; Kim et al., 2020; Wahed et al., 2022). Adversarial perturbations can undermine the reliability of SSL models, compromising their utility in real-world applications. One of the most effective strategies to address this challenge is **adversarial training**, which formulates the problem as a min-max optimization (Madry et al., 2017). Recent research has extended adversarial training to SSL frameworks (Kim et al., 2020; Moshavash et al., 2021; Wahed et al., 2022), demonstrating its effectiveness in enhancing robustness across methods such as SimCLR, Momentum Contrast (He et al., 2020), SwAV (Caron et al., 2020), and BYOL (Grill et al., 2020). However, these approaches face two important challenges: (i) **high computational costs** and (ii) **trade-offs** between clean accuracy and robustness.

To address these limitations, we introduce **Robust-EMP-SSL**, a framework building on Extreme-Multi-Patch Self-Supervised Learning (EMP-SSL) (Tong et al., 2023), which reduces training epochs by leveraging fixed-size image patches instead of traditional multi-scale cropping. Robust-EMP-SSL extends EMP-SSL to adversarial training scenarios by incorporating multiple crops per image to enhance data diversity, integrating invariance terms with regularization to prevent collapse, and optimizing training efficiency. Our contributions focus on the interplay between adversarial training and SSL, investigating key challenges and questions, including:

Table 1: **CF-AMC-SSL trains efficiently in fewer epochs, thereby reducing overall training time. By effectively employing multi-crop augmentations during base encoder training, it enhances both clean accuracy and robustness against PGD attacks.** Note that the highest values are indicated in red, while the second highest values are highlighted in blue.

| Models | CIFAR-10 | | | CIFAR-100 | | | Time | | |
|---|---|---|---|---|---|---|---|---|---|
| Base Encoder | Clean | PGD(4/255) | PGD(8/255) | Clean | PGD(4/255) | PGD(8/255) | (min/epoch) | Epochs | Total (min) |
| Patch-based EMP-SSL (baseline) 
 (16 patches, 5-step PGD, 30 epochs) | 61 | 37.65 | 16.95 | 39.26 | 14.38 | 4.22 | 17.67 | 30 | 530 |
| Crop-based EMP-SSL 
 (16 crops, 5-step PGD, 30 epochs) | **76.55** | **53.3** | **28.49** | **51.71** | **33.88** | **19.35** | 17.67 | 30 | 530 |
| Patch-based EMP-FreeAdv 
 (16 patches, m=3, 10 epochs) | 61.83 | 42.28 | 21.53 | 40.31 | 23.78 | 12.13 | 9.70 | 10 | **97** |
| Crop-based EMP-FreeAdv (CF-AMC-SSL) 
 (16 crops, m=3, 10 epochs) | **75.88** | **55.97** | **33.34** | **50.74** | **31.73** | **17.19** | 9.70 | 10 | **97** |
| Crop-based SimCLR (baseline) 
 (2 crops, 5-step PGD, 500 epochs) | 72.86 | 47.98 | 16.81 | 44.57 | 19.84 | 5.68 | 1.87 | 500 | 934 |
| Patch-based SimCLR 
 (2 patches, 5-step PGD, 500 epochs) | 65.44 | 41.85 | 17.19 | 43.71 | 21.87 | 8.33 | 1.87 | 500 | 934 |
| Crop-based SimCLR-FreeAdv 
 (2 crops, m=3, 167 epochs) | 70.25 | 48.34 | 24.5 | 47.64 | 26.53 | 11.7 | 0.94 | 167 | **157** |

1. **Can multiple crops or patches compensate for fewer training epochs in self-supervised adversarial training?** We examine whether increasing data diversity through multiple crops or patches can reduce computational overhead while maintaining performance.

2. **How does crop diversity affect the trade-off between clean accuracy and robustness?** By integrating the mechanisms of EMP-SSL with adversarial training, we analyze whether multiple crops can better balance the trade-off compared to SimCLR, which uses only two crops per image.

3. **What is the impact of the augmentation strategy (multi-scale crops vs. fixed-size patches) on robustness?** We investigate how the adoption of diverse cropping strategies affects the robustness of the model within the EMP-SSL framework.

4. **How do different evaluation strategies perform for adversarially trained models?** We compare robust multi-crop embedded aggregate (averaging embeddings from multiple crops) with the standard linear classifier applied to a single whole-image embedding.

5. **Can free adversarial training improve adversarial SSL?** We evaluate whether free adversarial training (Shafahi et al., 2019), known for its efficiency in supervised learning, can achieve competitive robustness in SSL under reduced training epochs.

We systematically compare EMP-SSL and SimCLR, a widely-used SSL method requiring hundreds of epochs to converge, using it as a baseline for its simplicity and ubiquity. Our key findings include:

- Increasing the number of multi-scale crops effectively compensates for fewer training epochs, enabling faster training without compromising performance (Table 1 row 2 versus rows 5 and 6).

- Robust-EMP-SSL with multi-scale crops achieves a better balance between clean accuracy and robustness than Robust-SimCLR used as the baseline (Table 1 row 2 versus row 5).

- Multi-scale crops within the Robust-EMP-SSL framework outperform fixed-size patches in adversarial SSL settings [1] (Table 1 row 2 versus row 1).

- Central cropping outperforms Multi-Crop Embedding Aggregation in terms of training time, clean accuracy, and robustness.

---

[1] Note that robust central crop evaluation is likely to be less effective in terms of accuracy with fixed-scale patch-based pretraining because the model lacks exposure to the entire image during pretraining. On the other hand, robust multi-patch evaluation is time-intensive, as it necessitates generating multiple adversarial examples per image for the adversarial training of the linear classifier.

- Free adversarial training (Shafahi et al., 2019) provides a cost-efficient solution for adversarial SSL, even under reduced training epochs (Table 1 rows 3, 4, and 7).

By addressing these questions, our work advances adversarial SSL by offering efficient and robust algorithms, paving the way for real-world applications with reduced computational costs and improved performance trade-offs.

## 2  Related Work

Several recent studies have advanced adversarially robust self-supervised learning. DeACL (Zhang et al., 2022) reformulates the problem into two sub-tasks—standard SSL for representation learning and pseudo-supervised adversarial training for robustness—achieving strong efficiency and robustness. ProFeAT (Addepalli et al., 2024) introduces feature-space adversarial objectives with a projection head to close the gap with supervised adversarial training. Adversarial Invariant Regularization (AIR) (Xu et al., 2023) applies causal invariance principles to improve both generalization and robustness, while broader adversarial self-supervised contrastive learning frameworks generate adversarial positives and negatives to strengthen contrastive pretraining. More recently, DAQ-SDP (Zhang et al., 2024) proposes diverse augmented queries and self-supervised double perturbation, showing that increasing augmentation diversity and applying perturbations at both input and model levels can significantly enhance robustness and transferability. In parallel, a positive mining approach (Kim et al., 2023) for unsupervised adversarial training has been proposed to address the ineffectiveness of untargeted adversarial examples in SSL, especially for non-contrastive frameworks. By selecting the most confusing yet semantically similar targets and perturbing instances toward them, this method generates more effective adversaries, yielding substantial robustness gains for non-contrastive SSL and consistent improvements for contrastive SSL. These methods propose complementary strategies that can be integrated into either SimCLR or EMP-SSL. In contrast, our work isolates the role of multi-crop versus pairwise crops under Free Adversarial Training to examine their direct impact on adversarial robustness and training efficiency.

## 3  Methodology

In this study, we focus on evaluating the robustness of Extreme-Multi-Patch Self-Supervised-Learning (EMP-SSL) (Tong et al., 2023), a method that employs multiple augmentations per image and achieves convergence in significantly fewer epochs compared to standard SSL methods. We selected SimCLR as the baseline for its simplicity and its standard SSL design, which uses two augmentations per image and requires hundreds of epochs for convergence. This makes SimCLR a suitable reference point to assess the efficiency and robustness of EMP-SSL.

### 3.1  Overview of SimCLR and EMP-SSL

#### 3.1.1  SimCLR: A Simple Framework for Contrastive Learning

SimCLR is a self-supervised learning framework designed to learn image representations by maximizing agreement between augmented views of the same image.

**Key Components:**

- **Data Augmentation:** Positive pairs are generated through random augmentations such as cropping, color jittering, and Gaussian blur. Each input image $\mathbf{x}$ produces two augmented views, $\hat{\mathbf{x}}_1$ and $\hat{\mathbf{x}}_2$.

- **Feature Extraction:** A backbone network $f(\cdot)$, typically a ResNet, maps each augmented view $\hat{\mathbf{x}}_i$ to a feature vector $\mathbf{h}_i$ in a high-dimensional space.

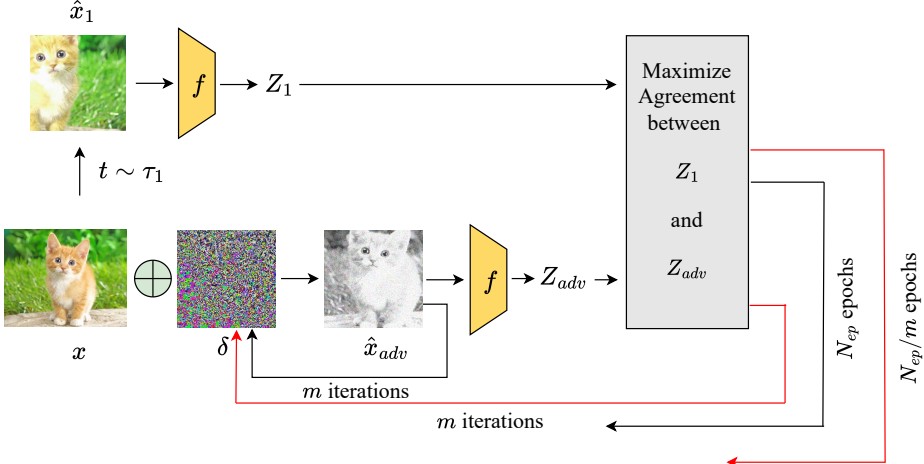

(a) The adversarially trained SimCLR vs. free adversarially trained SimCLR framework.

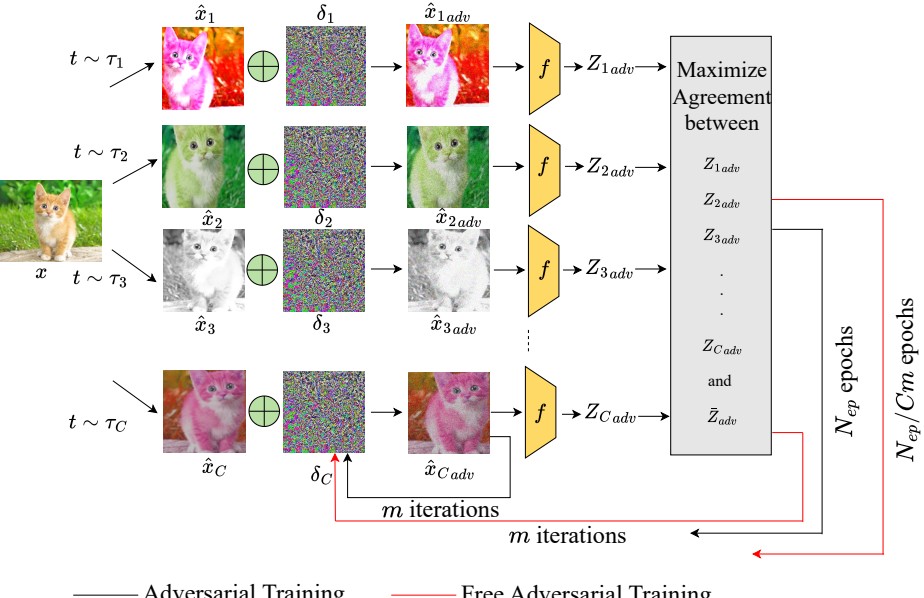

——— Adversarial Training     ——— Free Adversarial Training

(b) The adversarially trained crop-based EMP-SSL framework vs. the free adversarially trained, crop-based EMP-SSL (CF-AMC-SSL).

Figure 1: Illustration of workflow comparison.

- **Projection Head:** A multi-layer perceptron (MLP) $g(\cdot)$ projects the feature vector $\mathbf{h}_i$ into a lower-dimensional embedding space $\mathbf{z}_i$, where contrastive loss is applied:

$$\mathbf{z}_i = g(f(\hat{\mathbf{x}}_i)).$$

- **Contrastive Loss:** The loss function encourages embeddings of positive pairs to be similar while pushing apart embeddings of negative pairs. The contrastive loss is defined as:

$$l_{(i,j)} = -\log \frac{\exp(\text{sim}(\mathbf{z}_i, \mathbf{z}_j)/\tau)}{\sum_{k=1}^{2N} \mathbb{1}_{[k \neq i]} \exp(\text{sim}(\mathbf{z}_i, \mathbf{z}_k)/\tau)},$$

$$\mathcal{L}_{\text{contrastive}} = \frac{1}{2N} \sum_{k=1}^{N} \left[ l_{(2k-1,2k)} + l_{(2k,2k-1)} \right]$$

where:

- $\text{sim}(\mathbf{z}_i, \mathbf{z}_j)$ is the cosine similarity between embeddings,
- $\tau$ is a temperature parameter (scalar),
- $N$ is the number of images in a mini-batch (scalar).

SimCLR's framework efficiently captures meaningful representations by leveraging this contrastive learning approach.

### 3.1.2 EMP-SSL: Extreme-Multi-Patch Self-Supervised Learning

EMP-SSL extends the self-supervised learning paradigm by incorporating multiple fixed-scale patches per image and introducing novel loss terms to enhance representation consistency and generalization.

**Key Components:**

- **Multi-Patch Representation Learning:** During training, given a batch of $b$ images denoted as $\mathbf{X} = [\mathbf{x}_1, \ldots, \mathbf{x}_b]$, where $\mathbf{x}_j$ represents the $j$-th image in the batch, we first apply augmentations including dividing each image into $C$ fixed-scale patches to obtain $\mathbf{X}_1, \ldots, \mathbf{X}_C$, where $\mathbf{X}_i = [\mathbf{x}_1^i, \ldots, \mathbf{x}_b^i]$. The augmented image patches are then fed into the encoder to extract features $\mathbf{Z}_i = F(\mathbf{X}_i)$, which are subsequently concatenated into $\mathbf{Z} = [\mathbf{Z}_1, \ldots, \mathbf{Z}_C]$.

- **Invariance Term:** To encourage consistency among embeddings, an invariance term $D(\mathbf{Z}_i, \bar{\mathbf{Z}})$ aligns each patch's embedding $\mathbf{Z}_i$ with the average embedding $\bar{\mathbf{Z}}$:

$$\bar{\mathbf{Z}} = \frac{1}{C} \sum_{i=1}^{C} \mathbf{Z}_i, \quad D(\mathbf{Z}_i, \bar{\mathbf{Z}}) = \text{Tr}((\mathbf{Z}_i)^T \bar{\mathbf{Z}}).$$

- **Regularization Term:** A regularization term $R(\mathbf{Z})$ penalizes redundancy in feature embeddings by reducing correlations among dimensions:

$$R(\mathbf{Z}) = \frac{1}{2} \log \det \left( \mathbf{I} + \frac{d}{b\epsilon^2} \mathbf{Z}\mathbf{Z}^T \right),$$

where $\mathbf{Z}\mathbf{Z}^T$ is the covariance matrix of the embeddings, $b$ is the batch size (scalar), $\epsilon$ is a distortion size (scalar, $\epsilon > 0$), $d$ is the dimension of projection vectors (scalar), and $\mathbf{I}$ is the identity matrix.

- **Overall Objective Function:** EMP-SSL maximizes the sum of the invariance and regularization terms across all patches:

$$\mathcal{L}_{\text{EMP-SSL}} = \sum_{i=1}^{C} \left[ D(\mathbf{Z}_i, \bar{\mathbf{Z}}) + R(\mathbf{Z}_i) \right].$$

Using these innovative loss terms, EMP-SSL fosters stronger invariance and diversity in learned representations, making it a powerful approach to self-supervised learning.

### 3.2 Overview of Robust-SimCLR and Robust-EMP-SSL: Extending SimCLR and EMP-SSL with Adversarial Training

Initially, our analysis in Table 2 reveals that both the baseline EMP-SSL and SimCLR **without adversarial training** are vulnerable to adversarial attacks. This finding further underscores the lack of robustness in SimCLR, consistent with previous research findings (Ghofrani et al., 2023; Kim et al., 2020). We extend the SimCLR and EMP-SSL frameworks by incorporating adversarial training to improve robustness against adversarial attacks, as shown in Figure 1. Adversarial training integrates adversarial examples into the learning process, enabling models to better withstand perturbations. In the following, we describe the generation of adversarial examples and the modifications to the training objectives for both frameworks.

### 3.2.1 Robust-SimCLR: Adversarial Contrastive Learning

In the robust version of SimCLR, adversarial training enhances the model's resilience. For each image in the mini-batch, adversarial examples are generated using PGD attacks. Both the original augmented view and its adversarial counterpart are treated as positive pairs. The contrastive loss function is updated to maintain similarity between these pairs while distinguishing them from negative samples, thereby reinforcing the model's ability to generalize under adversarial perturbations.

### 3.2.2 Robust-EMP-SSL: Adversarial Multi-Patch Learning

In robust EMP-SSL, adversarial training is integrated to strengthen the model's defenses. The representation for each image is aggregated from multiple crops or patches, and adversarial perturbations are applied independently to each crop (patch).

**Adversarial Perturbation Process:**

1. The image is divided into multiple crops or patches.

2. An adversarial perturbation is generated and updated independently for each crop or patch.

3. Each crop is perturbed individually, rather than applying a shared perturbation across all crops.

This independent generation of adversarial examples results in more diverse and challenging perturbations, boosting the model's robustness.

**Updated Training Objective:**  The training objective includes:

- **Regularization Term** $R(\mathbf{Z}_{i,adv})$: Penalizes high correlations among adversarial representations.

- **Consistency Term** $D(\mathbf{Z}_{i,adv}, \bar{\mathbf{Z}}_{adv})$: Promotes consistency among adversarial embeddings.

These terms regularize adversarial representations while maintaining their similarity to the original augmented views, preventing overfitting to adversarial examples.

## 4 Experiments and Results

Building on our methodology, we investigate the performance of Robust-EMP-SSL under adversarial training and evaluate its trade-offs between clean accuracy, robustness, and computational efficiency. EMP-SSL significantly reduces the number of training epochs in self-supervised learning by increasing the number of fixed image patches augmented per image. This approach diverges from traditional methods like SimCLR, which rely on extensive training epochs and use only two augmented multi-scale crops per image. Through a comprehensive analysis of EMP-SSL within an adversarial training framework, we aim to understand the relationship between training epochs, image crop choices, model robustness, and accuracy.

**Empirical Study Objective:** These evaluation methods provide insights into the model's capabilities in terms of accuracy and robustness, incorporating both conventional and novel multi-patch and multi-crop embedding strategies.

**Threat Model:** To assess model robustness, we employ a threat model in which the adversary has full knowledge of the base encoder's architecture, network parameters, and those of the linear classifier. Adversarial attacks are generated end-to-end using the cross-entropy loss function.

### 4.1 Experiment Setup

Our experimental setup follows that of (Tong et al., 2023). We employ a ResNet-18 as the base encoder for all experiments unless otherwise specified. We train EMP-SSL models for 30 epochs and SimCLR models for

500 epochs, as EMP-SSL converges significantly faster. In all adversarial training scenarios, we use a 5-step PGD attack under the $l_\infty$ norm with a maximum perturbation limit of $\epsilon = 8/255$ unless stated otherwise. The models are evaluated against 20-step PGD attacks, and results are further validated using Auto-Attack (Croce & Hein, 2020). We report the top-1 test accuracy across all settings to evaluate robustness. After sampling the patches and crops, their resolution is adjusted to match the original image size before passing them through ResNet-18 for embedding. Since our goal is not to evaluate transfer learning (cross-dataset validation), both the base encoder and the linear classifier are trained on the same dataset in all experiments.

### 4.2 Evaluating Augmentation Strategies for Adversarial Training in Robust-EMP-SSL

To enhance the robustness of EMP-SSL under adversarial training, we analyze two augmentation strategies and their impact on representation diversity, robustness, and computational efficiency:

1. **Crop-Based Method:** In this approach, random crops are taken from the augmented image, with crop sizes ranging from $9 \times 9$ to $32 \times 32$ pixels. This method introduces a greater degree of spatial variability, requiring the model to learn more robust and generalizable representations. The key advantages of this method include:

   - **Enhanced Spatial Generalization:** Since crops are sampled from different regions of the image, the model learns invariant representations that are less dependent on specific spatial locations.
   - **Diversity in Adversarial Perturbations:** Adversarial examples generated on smaller cropped regions can vary significantly from those generated on full-image patches, forcing the model to handle adversarial attacks across multiple spatial scales.
   - **Improved Feature Learning:** The network is trained to extract meaningful features from both global and local perspectives, benefiting both clean accuracy and robustness.

   However, this method comes with added computational costs due to the need to generate multiple adversarial examples for diverse crops per image.

2. **Patch-Based Method:** Fixed-scale patches are extracted at predefined scales from the image, maintaining a structured and consistent learning framework. The benefits of this approach include:

   - **Simplicity:** Since patches are sampled at fixed locations and sizes, the method is simpler and more predictable during standard training, compared to random cropping.
   - **Stability in Representation Learning:** Fixed patches provide a stable learning process by reducing randomness. This can improve clean accuracy, especially when combined with multi-patch embedding aggregation during evaluation.

   However, this method may not achieve the same level of robustness as the crop-based approach because of its limited variability. Since adversarial training benefits from exposure to a wider range of perturbations, the lack of randomness in patch selection could result in less effective robustness improvements.

> **Justification for Evaluating Both Methods:** Given the inherent trade-off between diversity and efficiency, the evaluation of both augmentation strategies helps us determine the optimal balance between clean accuracy, robustness, and computational cost in adversarial SSL. The crop-based method offers superior robustness due to increased spatial diversity, whereas the patch-based method improves training simplicity and stability. By comparing these two approaches, we provide insights into how augmentation strategies impact adversarial training in self-supervised learning.

Our evaluation focuses on linear probing accuracy, considering the following two key evaluation strategies:

1. **Standard Central Crop Assessment:** This conventional method trains and evaluates a linear classifier using a single fixed central patch from each image, where the entire image serves as the central patch.

Table 2: **Comparative results of clean data performance and robustness against PGD attacks: baseline SimCLR versus EMP-SSL with standard pretraining on CIFAR10 and CIFAR100 datasets.**

| Models | | CIFAR-10 | | | | CIFAR-100 | | | |
|---|---|---|---|---|---|---|---|---|---|
| Linear Classifier | Base Encoder | Clean | PGD(4/255) | PGD(8/255) | PGD(16/255) | Clean | PGD(4/255) | PGD(8/255) | PGD(16/255) |
| Central Crop | SimCLR | 86.65 | 0.13 | 0 | 0 | 62.5 | 0.74 | 0.53 | 0.45 |
| | EMP-SSL | 75.02 | 0 | 0 | 0 | 44.31 | 0.02 | 0.02 | 0.02 |
| 32 Crops (Patches) | SimCLR | 86.68 | 0.02 | 0 | 0 | 65.21 | 0.18 | 0.12 | 0.07 |
| | EMP-SSL | 92.85 | 0.04 | 0.01 | 0 | 71.82 | 0.46 | 0.15 | 0.08 |
| 64 Crops (Patches) | SimCLR | 89.31 | 0.01 | 0 | 0 | 66.3 | 0.17 | 0.12 | 0.12 |
| | EMP-SSL | 93.29 | 0.02 | 0.03 | 0.01 | 72.3 | 0.5 | 0.2 | 0.09 |

2. **Multi-Patch (Multi-Crop) Embedding Aggregation Evaluation:** In contrast to the standard central crop assessment, this method constructs an image embedding by inputting a specified number of crops (patches) into the base encoder. These crop (patch) embeddings are then combined by averaging and fed into the linear classifier. Henceforth, we refer to this evaluation method as the **"$n$ Crops (Patches)"** linear classifier. Note that patches (crops) are sampled with the same scale factor as during the pre-training phase.

### 4.3 Both SimCLR and EMP-SSL Are Vulnerable to Adversarial Attacks Under Standard Training

We first examine the vulnerability of standard SimCLR and EMP-SSL training to adversarial attacks. The goal of this experiment is twofold: (1) to assess whether employing multiple fixed-sized patches in SSL (as in EMP-SSL) inherently enhances robustness compared to methods like SimCLR, which use only one pair of differently scaled crops, and (2) to evaluate whether multi-patch (multi-crop) aggregated evaluation improves robustness over standard central crop evaluation.

Detailed results for clean accuracy and adversarial robustness under PGD attacks are presented in Table 2 for CIFAR-10 and CIFAR-100.

> The findings highlight that while multi-patch aggregated evaluation enhances performance on clean data, both SimCLR and EMP-SSL remain highly vulnerable to adversarial attacks, indicating the necessity of adversarial training.

### 4.4 Robust Crop-Based EMP-SSL Improves Both Clean Accuracy and Robustness Compared to Robust SimCLR

We next apply adversarial training to both SimCLR and EMP-SSL, evaluating their performance primarily through central cropping, which balances computational efficiency with accuracy. Figure 2 presents results across different training configurations, highlighting the following key findings:

- Training SimCLR with only two augmentations per image results in a significant trade-off between clean accuracy and robustness.

- Crop-based EMP-SSL exhibits higher robustness against adversarial attacks, whereas patch-based EMP-SSL achieves superior clean accuracy under multi-patch aggregation evaluation (detailed in the Appendix A.1).

- Central cropping provides an efficient evaluation method while maintaining strong clean accuracy and robustness (detailed in the Appendix A.1).

To further understand these trade-offs, we conducted additional ablation studies, examining the effects of different numbers of crops and patches on model performance. Detailed findings are provided in the Appendix A.2, which show that

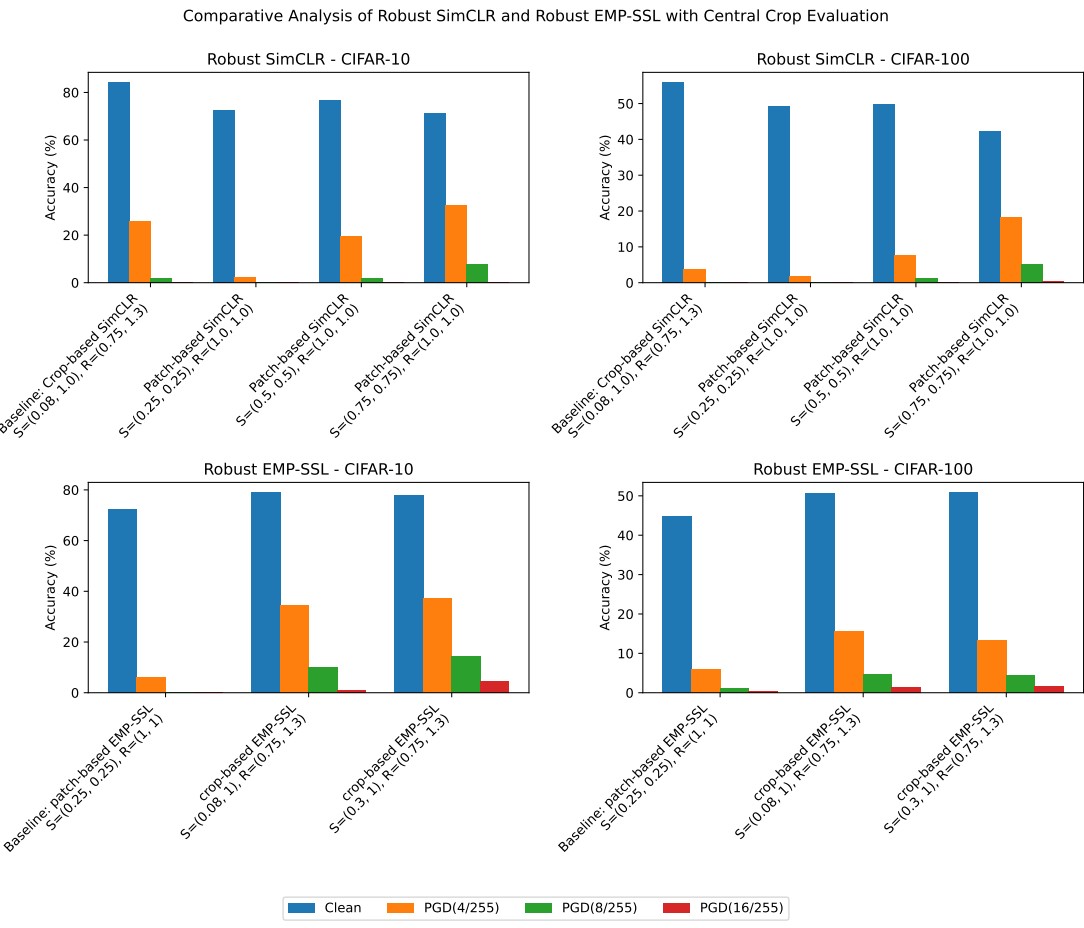

Figure 2: **Evaluation of robustness against PGD attacks through adversarial pretraining on CIFAR-10 and CIFAR-100 datasets.** We compare the performance of robust SimCLR and robust EMP-SSL with central crop evaluation under different training configurations. Our analysis includes the evaluation of patch-based SimCLR with varying patch sizes and baseline SimCLR, revealing a noticeable trade-off between clean accuracy and robustness. Larger patch sizes in robust SimCLR improve robustness but reduce clean accuracy. Additionally, we compare crop-based EMP-SSL (with varying crop sizes) to baseline EMP-SSL, demonstrating that the crop-based approach significantly enhances robustness. Notably, Robust EMP-SSL achieves a superior balance between clean accuracy and robustness compared to robust SimCLR. The variables $S$ and $R$ correspond to the scales and ratios used in the PyTorch framework's RandomResizedCrop method.

Increasing the number of patches generally enhances clean accuracy under multi-patch aggregation evaluation, while a well-calibrated number of crops balances robustness and accuracy effectively. These insights reinforce the effectiveness of crop-based EMP-SSL in adversarial settings while also highlighting the practical considerations of augmentation strategies in robust self-supervised learning.

## 4.5 Robust Crop-Based EMP-SSL with Robust Linear Evaluation

While robust crop-based EMP-SSL demonstrates improved robustness in standard evaluation, it is crucial to further scrutinize its adversarial resilience in downstream applications. To this end, we employ an additional assessment known as robust linear evaluation (r-LE) (Kim et al., 2020). This evaluation aligns with the broader objectives of our study, as it provides deeper insights into the transferability of adversarial robustness learned during self-supervised training and its impact on downstream tasks.

Table 3: **Evaluation of Robust EMP-SSL and Robust SimCLR across different adversarial training scenarios on CIFAR-10 and CIFAR-100 datasets:** The findings imply that boosting the robustness of the linear classifier contributes to enhancing the overall robustness of both the robust SimCLR and EMP-SSL.

| Models | | CIFAR-10 | | | | CIFAR-100 | | | |
|---|---|---|---|---|---|---|---|---|---|
| Linear Classifier | Base Encoder | Clean | PGD(4/255) | PGD(8/255) | PGD(16/255) | Clean | PGD(4/255) | PGD(8/255) | PGD(16/255) |
| Central Crop | Robust SimCLR | **84.24** | 25.68 | 1.97 | 0.07 | **55.91** | 3.58 | 0.18 | 0 |
| | Robust EMP-SSL Crop-based(16) | 80.72 | 33.62 | 8.95 | 0.92 | 51.83 | 19.3 | 6.85 | 1.73 |
| Robust Central Crop (r-LE) | Robust SimCLR | 72.86 | 47.98 | 16.81 | 0.33 | 44.57 | 19.84 | 5.68 | 0.26 |
| | Robust EMP-SSL Crop-based(16) | 76.55 | **53.3** | **28.49** | **3.96** | 51.71 | **33.88** | **19.35** | **4.92** |

The r-LE approach consists of two stages: (1) pretraining the base encoder using the robust crop-based EMP-SSL algorithm and (2) freezing the base encoder while separately adversarially training a linear classifier on top of it. This setup allows us to decouple the robustness contributions of the encoder and the linear classifier, revealing whether adversarial robustness can be preserved and enhanced in a transfer learning setting.

The results, presented in Table 3, highlight the following key observations:

- Adversarially training the linear classifier improves overall robustness, demonstrating that robust features learned via EMP-SSL can be further refined to withstand stronger attacks.

- Robust crop-based EMP-SSL maintains a superior balance between clean accuracy and robustness compared to robust SimCLR, reinforcing its effectiveness in adversarial self-supervised learning.

- The effectiveness of r-LE suggests that adversarial robustness in self-supervised representations is not solely determined by the base encoder but can also be enhanced at the linear classifier stage, improving overall robustness.

By incorporating r-LE into our evaluation framework, we establish a more comprehensive understanding of the adversarial robustness of self-supervised representations and validate the effectiveness of robust crop-based EMP-SSL in both feature learning and transferability.

## 4.6 Cost-Free Adversarial Multi-Crop Self-Supervised Learning Evaluation

Inspired by our findings on robust EMP-SSL, we introduce a novel adversarial self-supervised learning method that achieves convergence in fewer than 10 epochs. We apply free adversarial training (Shafahi et al., 2019) to the crop-based EMP-SSL framework (see Figure 1), referring to it as **Cost-Free Adversarial Multi-Crop**

---

**Algorithm 1:** CF-AMC-SSL learning algorithm for $N_{ep}$ epochs, given some radius $\epsilon$, $m$ minibatch replays, $C$ number of crops, and a dataset of size $D$ for an encoder $f_\theta$ and a projector $g_\theta$

---

**Initialize:** $\delta \leftarrow 0$

1 //Iterate $N_{ep}/m$ times to account for minibatch replays and run for $N_{ep}$ total epochs
2 **for** epoch = 1 to $N_{ep}/m$ **do**
3    **for** $i = 1$ to $D$ **do**
4       //Augment data for CF-AMC-SSL learning:
5       **for** $k = 1$ to $C$ **do**
6          Draw augmentation function $t_k$;
7          $\hat{x}_{i,k} = t_k(x_i)$;
8       **end**
9       **for** $j = 1$ to $m$ **do**
10          //Compute gradients for perturbation and model weights simultaneously:
11          $\nabla\delta, \nabla\theta = \nabla\mathcal{L}_{\mathcal{EMP-SSL}}(f \circ g_\theta(\hat{x}_{i,k} + \delta))$
12          $\delta = \delta + \epsilon \cdot sign(\nabla\delta)$ //Update $\delta$ with the gradients calculated
13          $\delta = \max(\min(\delta, \epsilon), -\epsilon)$
14          $\theta = \theta - \nabla\theta$ // Update model weights with some optimizer
15       **end**
16    **end**
17 **end**

---

Table 4: **Evaluation of CF-AMC-SSL and SimCLR-FreeAdv algorithms:** The results show that CF-AMC-SSL trains efficiently in fewer epochs, reducing the overall training time. Additionally, employing multi-crop augmentations in CF-AMC-SSL during base encoder training effectively improves both accuracy and robustness. Note that the topmost and the second-highest values are indicated in red and blue, respectively.

| Models | | CIFAR-10 | | | | CIFAR-100 | | | | Time | ImageNet-100 | | | |
|---|---|---|---|---|---|---|---|---|---|---|---|---|---|---|
| Linear Classifier | Base Encoder ResNet-18 | Clean | PGD 4/255 | PGD 8/255 | PGD 16/255 | Clean | PGD 4/255 | PGD 8/255 | PGD 16/255 | (min) | Clean | PGD 4/255 | PGD 8/255 | PGD 16/255 |
| | Crop-based SimCLR (5-step PGD, 500 epochs) | 72.86 | 47.98 | 16.81 | 3.3 | 44.58 | 19.84 | 5.68 | 0.26 | 934 | 49.64 | 30.84 | 15.3 | 1.94 |
| | SimCLR-FreeAdv (m=3, 167 epochs) | 70.25 | 48.34 | 24.5 | 2 | 47.64 | 26.53 | 11.7 | 1.26 | 157 | 30.26 | 16.14 | 6.06 | 0.25 |
| | SimCLR-FreeAdv (m=5, 100 epochs) | 69.97 | 51.36 | 30.84 | 5.7 | 45.69 | 29.43 | 16.15 | 3.1 | 157 | 25.26 | 14.12 | 5.88 | 0.7 |
| Robust Central Crop | Crop-based EMP-SSL (16 crops, 5-step PGD, 30 epochs) | 76.55 | 53.3 | 28.49 | 3.96 | 51.71 | 33.88 | 19.35 | 4.92 | 530 | 50.04 | 26.54 | 10.2 | 0.5 |
| | CF-AMC-SSL (16 crops, m=3, 10 epochs) | 75.78 | 55.97 | 33.34 | 6.24 | 50.74 | 31.73 | 17.19 | 3.49 | 97 | 40.22 | 21.24 | 8.12 | 1.14 |
| | CF-AMC-SSL (16 crops, m=5, 6 epochs) | 71.89 | 54.2 | 34.94 | 8.55 | 45.84 | 30.1 | 17.84 | 5.15 | 97 | 34.38 | 18.82 | 8.22 | 1.2 |
| | CF-AMC-SSL (16 crops, m=5, 10 epochs) | - | - | - | - | - | - | - | - | - | 46.26 | 27.86 | 13.94 | 2.16 |
| | CF-AMC-SSL (16 crops, m=5, **18 epochs**) | 76.28 | 58.06 | 37.5 | 9.39 | 52.01 | 33.3 | 19.34 | 5.06 | 291 | – | – | – | – |
| | CF-AMC-SSL (16 crops, **m=12**, 10 epochs) | 55.84 | 43 | 30.84 | **12.4** | 31.33 | 21.86 | 14.62 | **6.14** | 388 | – | – | – | – |
| | Supervised-FreeAdv (m=3, 300 epochs) | 82.63 | 47.12 | 16.27 | 1.3 | 52.07 | 20.2 | 6.34 | 0.92 | 155 | – | – | – | – |
| | Supervised-FreeAdv (m=7, 300 epochs) | 74.63 | 48.56 | 23.75 | 2.87 | 39.88 | 19.97 | 8.14 | 1.12 | 360 | – | – | – | – |

**Self-Supervised Learning (CF-AMC-SSL)**. This approach significantly reduces the computational burden of adversarial SSL training, decreasing the required epochs by nearly two orders of magnitude (Algorithm 1).

Our method employs an iterative adversarial training strategy that integrates multi-crop augmentations. Specifically, it repeats each training iteration $m$ times within a minibatch, reusing gradient information from previous updates. This enables efficient adversarial example generation before progressing to the next iteration. Experimental results for various values of $m$ are presented in Table 4, alongside comparisons to the

Table 5: **Evaluation of Different Learning Algorithms Using ResNet-50 as the Base Encoder:** This experiment demonstrates the generalizability of our findings when employing a larger base encoder, such as ResNet-50. Additionally, it highlights that increasing the number of iterations for PGD enhances the model's robustness against larger perturbations.

| Models | | CIFAR-10 | | | | CIFAR-100 | | | |
|---|---|---|---|---|---|---|---|---|---|
| Linear Classifier | Base Encoder ResNet-50 | Clean | PGD(4/255) | PGD(8/255) | PGD(16/255) | Clean | PGD(4/255) | PGD(8/255) | PGD(16/255) |
| Robust Central Crop | CF-AMC-SSL (16 crops, m=7, 9 epochs) | **73.43** | **56.81** | **38.31** | 11.03 | **47.4** | **32.19** | **19.9** | 6.21 |
| | CF-AMC-SSL (16 crops, m=3, 10 epochs) | **75.89** | **57.61** | **35.19** | 6.51 | **53.34** | **33.08** | **18.15** | 3.72 |
| | SimCLR-FreeAdv (m=7, 150 epochs) | 49.49 | 39.09 | 28.88 | **12.76** | 25.49 | 18.36 | 12.51 | 5.51 |
| | SimCLR-FreeAdv (m=3, 150 epochs) | 66.89 | 47.52 | 26.9 | 3.68 | 38.95 | 24.14 | 12.17 | 1.61 |
| | VICReg-FreeAdv (m=7, 150 epochs) | 56.71 | 44.04 | 31.84 | 11.58 | 27.81 | 20.71 | 14.79 | **6.44** |
| | VICReg-FreeAdv (m=3, 150 epochs) | 71.25 | 53.06 | 32.64 | **6.72** | 45.35 | 30.35 | 17.13 | **4.29** |

Table 6: **Evaluation of different learning algorithms against AutoAttack (AA):** The Autoattack evaluation confirms that using multi-crop augmentations in CF-AMC-SSL during base encoder training improves both accuracy and robustness effectively. Note that the topmost and the second-highest values are indicated in red and blue, respectively.

| Models | | CIFAR-10 | | | | CIFAR-100 | | | |
|---|---|---|---|---|---|---|---|---|---|
| Linear Classifier | Base Encoder ResNet-18 | Clean | AA(4/255) | AA(8/255) | AA(16/255) | Clean | AA(4/255) | AA(8/255) | AA(16/255) |
| Robust Central Crop | Crop-based EMP-SSL (16 crops, 5-step PGD, 30 epochs) | 76.55 | 23.93 | 26.57 | 7.81 | 51.71 | 33.88 | 19.35 | 4.92 |
| | CF-AMC-SSL (16 crops, m=3, 10 epochs) | 75.78 | 51.52 | 27.74 | 6.59 | 50.74 | 27.05 | 15.35 | 5.23 |
| | CF-AMC-SSL (16 crops, m=5, 6 epochs) | 71.89 | 50.76 | 30.14 | 8.44 | 45.84 | 26.99 | 16.3 | 5.83 |
| | Crop-based SimCLR (5-step PGD, 500 epochs) | 72.86 | 16.66 | 12.57 | 10.5 | 44.58 | 8.81 | 6.21 | 5.29 |
| | SimCLR-FreeAdv (m=3, 167 epochs) | 70.25 | 46.62 | 22.31 | 4.76 | 47.64 | 23.35 | 13.6 | 3.93 |
| | SimCLR-FreeAdv (m=5, 100 epochs) | 69.97 | 48.91 | 27.51 | 6.32 | 45.69 | 25.39 | 14.28 | 5.12 |

conventional 5-step PGD adversarially trained crop-based EMP-SSL model, which requires approximately five times the training time of CF-AMC-SSL variants.

For a broader comparison, we extend free adversarial training to SimCLR and its supervised counterpart (termed SimCLR-FreeAdv and Supervised-FreeAdv, respectively), with results summarized in Table 4. All experiments were conducted on a single RTX A6000 GPU, and runtime evaluations were performed on CIFAR-10 and CIFAR-100 datasets. Although multi-crop augmentations in joint-embedding SSL may initially appear computationally expensive, our findings reveal significant efficiency gains.

Robust EMP-SSL converges in 30 epochs—significantly fewer than the 500 epochs required for robust Sim-CLR—and achieves faster runtime (530 minutes vs. 934 minutes). CF-AMC-SSL further improves efficiency, leveraging free adversarial training (Shafahi et al., 2019) to reduce runtime to just 97 minutes—over five times faster than robust EMP-SSL—while maintaining comparable performance. Further details on computational efficiency are provided in Appendix A.3. Additionally, CF-AMC-SSL achieves a superior trade-off between clean accuracy and adversarial robustness, outperforming both robust SimCLR and its supervised variant even when label information is available.

To assess generalizability, we extended our experiments by increasing the number of Free-AT steps $m$ and training epochs. We observed that higher values of $m$ improve robustness, particularly under stronger perturbations (e.g., $\epsilon = 16/255$). We further evaluated CF-AMC-SSL on the ImageNet-100 dataset, where

the results followed a similar trend, reinforcing the efficiency and robustness of our approach. Additionally, we tested our method using ResNet-50 as the base encoder (Table 5), confirming that employing a larger architecture and increasing the number of PGD iterations enhances robustness, especially against stronger attacks. To offer a point of comparison with frameworks that rely solely on positive pairs, we also evaluated VICReg under the same setup. Finally, the Appendix A.4 reports robustness evaluations under additional adversarial attacks, including the Fast Gradient Sign Method (FGSM) (Goodfellow et al., 2014), the Carlini–Wagner (CW) attack (Carlini & Wagner, 2017) with 100 optimization steps, and the gradient-free Square Attack (Andriushchenko et al., 2020). These evaluations offer a more comprehensive assessment of robustness beyond PGD, further demonstrating the broader effectiveness of CF-AMC-SSL under diverse threat models.

In summary, our framework provides two significant advantages:

1. **Efficient training:** CF-AMC-SSL dramatically reduces training time, enabling adversarial self-supervised learning with far fewer epochs.

2. **Enhanced accuracy and robustness:** Multi-crop augmentations during base encoder training significantly improve both clean accuracy and adversarial robustness.

Finally, to further validate our approach, we evaluated CF-AMC-SSL against Auto-Attack (Table 6), confirming the reliability of our findings.

## 5 Discussion

The empirical evaluation of CF-AMC-SSL provides key insights into its effectiveness in improving adversarial self-supervised learning. Our results demonstrate that CF-AMC-SSL successfully enhances robustness while maintaining competitive clean accuracy, significantly outperforming robust SimCLR in terms of efficiency and adversarial resilience. By integrating free adversarial training with crop-based EMP-SSL, we enable substantial reductions in training time while preserving the ability to learn robust representations.

Our evaluation results indicate that leveraging a moderate number of augmentations per image during training enhances adversarial robustness, as the model encounters a broader range of perturbed images. This allows it to learn representations that emphasize content over style variations (e.g., color, textures, and background), ultimately leading to improved robustness. The empirical findings further highlight the impact of augmentation diversity, confirming that CF-AMC-SSL effectively balances robustness and efficiency through multi-crop self-supervised learning.

Similar to prior studies (Chen et al., 2022; Li et al., 2022), our method defines the representation of an image $x$ as the average of embeddings $h_1, \ldots, h_n$ across all its crops (a bag-of-words approach). Our results show that these learned representations are more robust than traditional representations based on full-image embeddings. Furthermore, selecting $n$ greater than two accelerates the learning of co-occurrence patterns across crops, reinforcing the efficiency of CF-AMC-SSL in adversarial settings.

Additionally, we demonstrate that free adversarial training can be successfully integrated into self-supervised learning, independent of the specific loss function used. Even with significantly fewer training epochs, free adversarial training preserves robustness, establishing CF-AMC-SSL as a scalable and efficient adversarial SSL method.

From an adversarial robustness perspective, CF-AMC-SSL employs a comprehensive training objective that includes both a regularization term $R(\mathbf{Z}_{i,adv})$ and an invariance term $D(\mathbf{Z}_{i,adv}, \bar{\mathbf{Z}}_{adv})$. The regularization term enforces constraints on adversarial representations to enhance robustness, while the invariance term ensures that adversarial perturbations remain consistent with their average representation $\bar{\mathbf{Z}}_{adv}$. This clustering effect mitigates extreme adversarial distortions, resulting in more stable and generalizable feature representations. Consequently, CF-AMC-SSL achieves a strong balance between robustness and clean accuracy, maintaining performance on clean data while enhancing resistance to adversarial perturbations.

### 5.1 Empirical Observations and Insights

Our empirical analysis yields the following key insights:

- **Trade-Off Between Clean Accuracy and Robustness:** Increasing the number of crops enhances robustness, but excessive cropping can lead to diminishing returns and slight reductions in clean accuracy. The ablation studies in the appendices further confirm optimal crop configurations for maintaining this trade-off.

- **Impact of Augmentation Strategy:** Multi-crop augmentation improves generalization under adversarial attacks compared to traditional contrastive methods. However, our findings reveal that crop-based EMP-SSL is particularly effective in preserving adversarial robustness, while patch-based EMP-SSL achieves superior clean accuracy under multi-patch aggregation evaluation.

- **Efficiency of Free Adversarial Training:** Our results validate that integrating free adversarial training into crop-based EMP-SSL significantly accelerates training without sacrificing robustness. This makes it an ideal solution for large-scale self-supervised learning tasks.

- **Generalization to Different Architectures and Datasets:** The results on ImageNet-100 and ResNet-50 confirm that CF-AMC-SSL extends effectively beyond CIFAR datasets, demonstrating its scalability across different model architectures and data distributions.

- **Evaluation Under Stronger Attacks:** Our results against Auto-Attack, the Carlini–Wagner (CW) attack, and the gradient-free Square Attack demonstrate that CF-AMC-SSL maintains robustness under stronger adversarial scenarios, further confirming its reliability in more challenging threat settings.

These observations provide a deeper understanding of the efficiency, robustness, and applicability of CF-AMC-SSL, further validating its potential for real-world adversarial self-supervised learning tasks.

## 6 Conclusion

In this work, we conducted a comprehensive study on the adversarial robustness of Extreme-Multi-Patch Self-Supervised Learning (EMP-SSL) and proposed Cost-Free Adversarial Multi-Crop Self-Supervised Learning (CF-AMC-SSL) as an efficient solution for adversarial self-supervised learning. Our empirical evaluations demonstrated that increasing the number of multi-scale crops in adversarial training enhances model robustness while maintaining competitive clean accuracy. Unlike robust SimCLR, which relies on only a pair of crops per image and requires extensive training epochs, robust EMP-SSL achieves a superior balance between accuracy and robustness with significantly fewer training iterations.

Furthermore, we explored the integration of free adversarial training into self-supervised learning, leading to the development of CF-AMC-SSL. This method dramatically reduces the number of required training epochs—by nearly two orders of magnitude—without compromising performance. Our findings validate that free adversarial training is not only applicable to supervised learning but also highly effective in self-supervised settings, significantly improving training efficiency and model resilience to adversarial perturbations.

The empirical analysis highlighted several key insights: multi-crop augmentation enhances robustness, crop-based EMP-SSL performs better in adversarial settings compared to patch-based EMP-SSL, and CF-AMC-SSL achieves a strong balance between computational efficiency, clean accuracy, and robustness. Additionally, we demonstrated that CF-AMC-SSL generalizes effectively to different architectures and datasets, reinforcing its practical applicability.

Overall, our study contributes to the advancement of adversarial self-supervised learning by providing a scalable and efficient training strategy. The insights derived from this work pave the way for future research in optimizing adversarial self-supervised learning methodologies, making them more suitable for real-world deployment in security-critical applications.

## Acknowledgments

This work was partially supported by the National Science Foundation (Awards 2007202, 2107463, 2038080, and 2233873).

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

# A   Appendix

## A.1   Evaluation with 32- and 64-Patch Aggregation

In addition to central cropping, we evaluated the robust base encoders using multi-patch aggregation with 32 and 64 patches. These methods involve aggregating embeddings from multiple fixed-size patches during evaluation. While this approach provides insights into the robustness of the learned representations, it is computationally more intensive than central cropping.

Key observation from the results (Figures 3 and 4) is as follows:

- *Multi-patch aggregation enhances clean accuracy, especially when a larger number of patches (e.g., 64) is used. However, the computational cost increases significantly, making central cropping more practical for resource-constrained settings.*

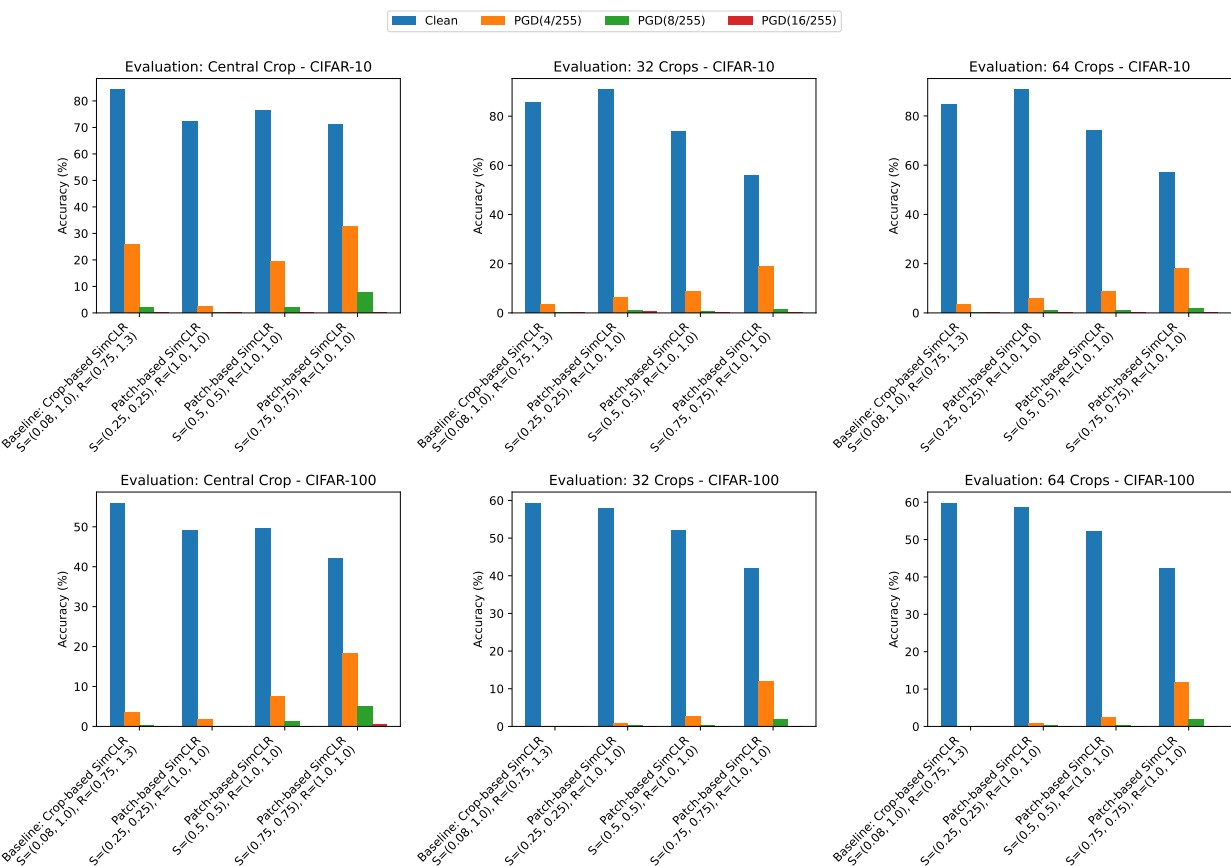

Figure 3: **Evaluating the robustness against PGD attacks through adversarial pretraining on CIFAR-10 and CIFAR-100 datasets, we compare the performance of patch-based SimCLR (with various patch sizes) to that of baseline SimCLR.** Our findings reveal a noticeable trade-off between clean accuracy and robustness. In addition, central cropping (first column) demonstrates higher efficiency in terms of overall complexity, clean accuracy, and robustness. Moreover, increasing patch sizes reduces clean accuracy but improves model robustness. Note that the variables $S$ and $R$ correspond to the scales and ratios employed in the PyTorch framework's RandomResizedCrop method.

## A.2 Ablation Study of Robust EMP-SSL

The ablation study analyzed the impact of varying the number of patches (crops) used for adversarial training in the EMP-SSL framework. The results for CIFAR10 and CIFAR100 are shown in Figure 5.

Findings include:

- *Increasing the number of patches during evaluation consistently improves clean accuracy in patch-based EMP-SSL.*

- *Moderate numbers of crops (e.g., 16) with crop-based EMP-SSL preserve a better trade-off between clean accuracy and robustness when evaluated using central cropping.*

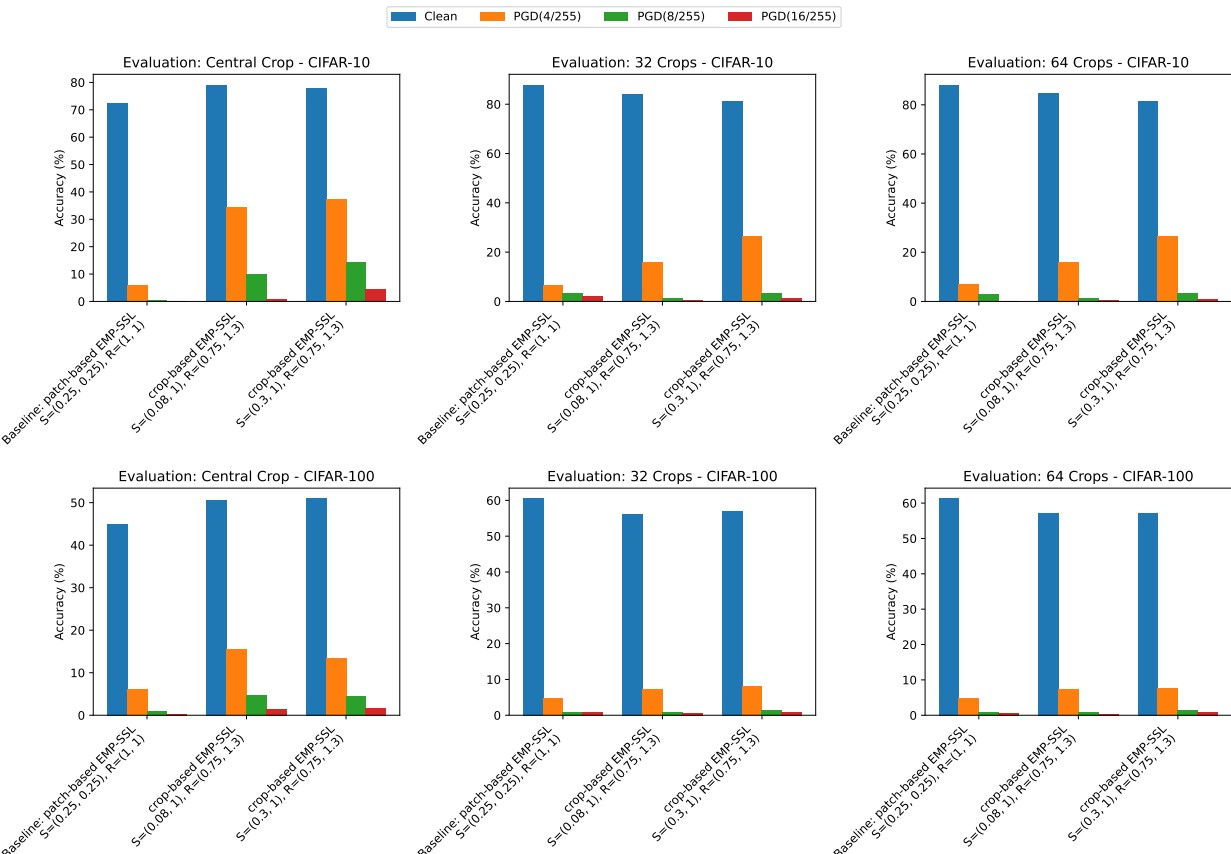

Figure 4: **Evaluating the robustness against PGD attacks through adversarial pretraining on CIFAR-10 and CIFAR-100 datasets, we compare the performance of crop-based EMP-SSL (with various crop sizes) to that of baseline EMP-SSL.** Our analysis reveals that the crop-based approach in EMP-SSL demonstrates enhanced robustness. Compared to the results presented in Figure 3, it is clear that Robust EMP-SSL achieves a superior balance between clean accuracy and robustness, in contrast to robust SimCLR. Here, the variables $s$ and $r$ denote the scales and ratios utilized for the RandomResizedCrop method within the PyTorch framework.

## A.3 Computational Efficiency Analysis

In this section, we provide a detailed analysis of the computational cost of all compared methods. Specifically, we disentangle *per-epoch* cost from *total* training cost and quantify the overhead introduced by adversarial training schemes and multiple views per image.

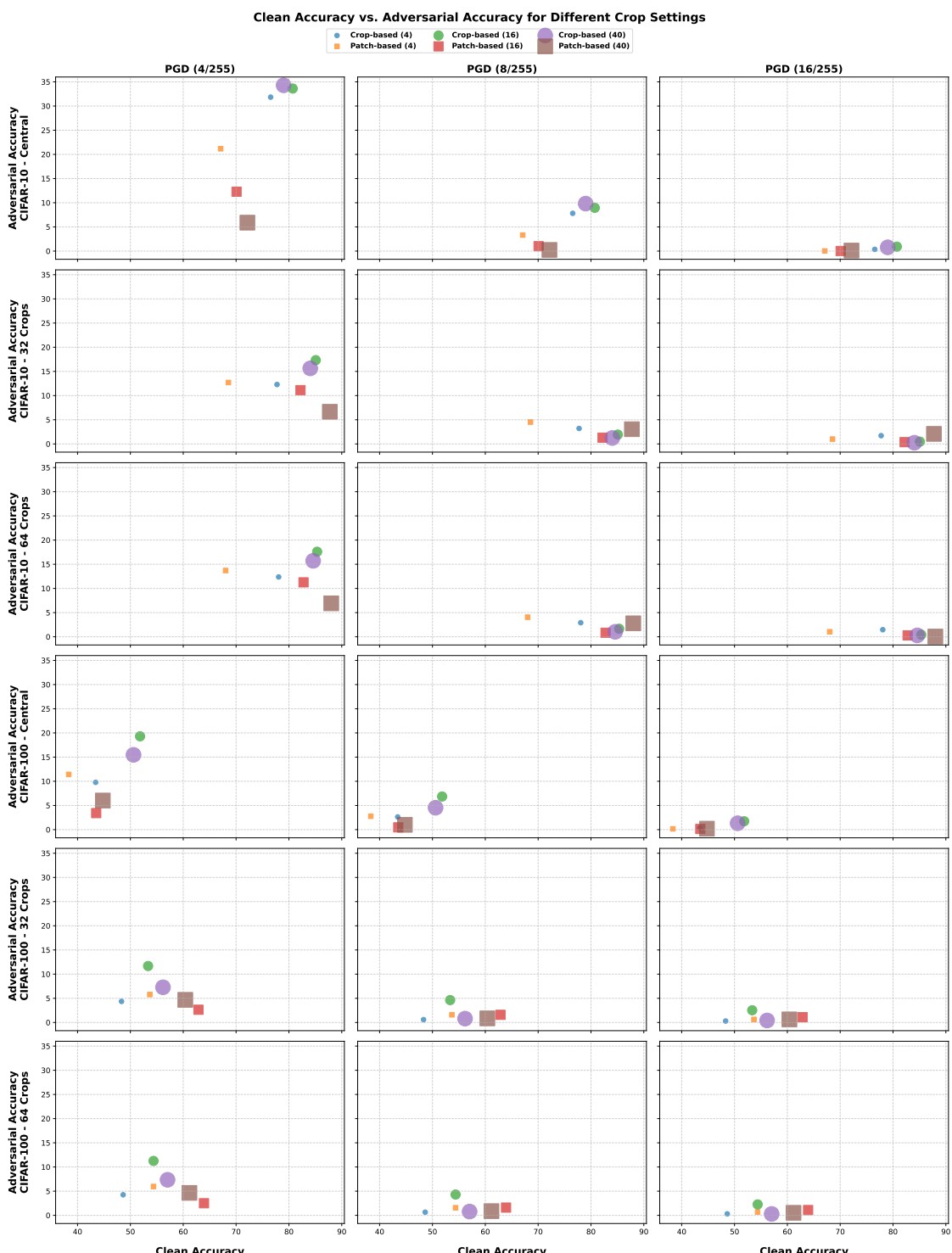

Figure 5: **Evaluation of robust EMP-SSL across different patch (crop) sizes on CIFAR-10 and CIFAR-100 datasets:** Our results emphasize that, when employing the patch-based EMP-SSL method with multi-patch aggregation during evaluation, a significant augmentation in the number of patches leads to a noticeable enhancement in clean accuracy. Furthermore, when using crop-based EMP-SSL with central-crop assessment, a more equitable balance between clean accuracy and model robustness can be established, especially evident with a moderate number of crops, such as 16. Note that "Crop-based (4)" means augmentation with scales (S) of (0.08, 1.0) and ratios (R) of (0.75, 1.3), with (4) denoting the number of crops. Similarly, "Patch-based (4)" involves scales (S) of (0.25, 0.25) and ratios (R) of (1.0, 1.0), with (4) representing the number of patches.

Table 7: **Compute ablation: GFLOPs and adversarial overhead.** Backbone is ResNet-18 (4.1 GFLOPs per forward pass). We report the number of views per image, adversarial scheme, the overhead multiplier relative to standard training, and the resulting effective compute per image per epoch and in total.

| Method | GFLOPs (per fwd) | Views (per img) | Adv. scheme | Overhead mult. | Eff. GFLOPs/epoch (per img) | Total GFLOPs (per img) |
|---|---|---|---|---|---|---|
| Patch-based EMP-SSL (baseline) 
 (16 patches, 5-step PGD, 30 epochs) | 4.1 | 16 | PGD-5 | 5.0 | 328.00 | 9,840.00 |
| Crop-based EMP-SSL 
 (16 crops, 5-step PGD, 30 epochs) | 4.1 | 16 | PGD-5 | 5.0 | 328.00 | 9,840.00 |
| Patch-based EMP-FreeAdv 
 (16 patches, m=3, 10 epochs) | 4.1 | 16 | Free-AT ($m=3$) | 1.3 | 85.28 | 852.80 |
| Crop-based EMP-FreeAdv (CF-AMC-SSL) 
 (16 crops, m=3, 10 epochs) | 4.1 | 16 | Free-AT ($m=3$) | 1.3 | **85.28** | **852.80** |
| Crop-based SimCLR (baseline) 
 (2 crops, 5-step PGD, 500 epochs) | 4.1 | 2 | PGD-5 | 5.0 | 41.00 | 20,500.00 |
| Patch-based SimCLR 
 (2 patches, 5-step PGD, 500 epochs) | 4.1 | 2 | PGD-5 | 5.0 | 41.00 | 20,500.00 |
| Crop-based SimCLR-FreeAdv 
 (2 crops, m=3, 167 epochs) | 4.1 | 2 | Free-AT ($m=3$) | 1.3 | 10.66 | 1,780.22 |

**Assumptions.**

1. Backbone is ResNet-18 with 4.1 GFLOPs per forward pass (identical across methods).

2. **Views** indicates the number of crops or patches per image used during pretraining.

3. **Overhead multiplier** is relative to standard (non-adversarial) training:
   - PGD-5 requires approximately $5\times$ compute overhead due to five gradient evaluations per batch.
   - Free-AT with $m=3$ reuses gradients across mini-batches, resulting in an amortized overhead of approximately $1.3\times$.

4. **Eff. GFLOPs/epoch** is computed as:

$$\text{Eff. GFLOPs/epoch} = \text{GFLOPs} \times \text{Views} \times \text{Overhead}.$$

5. **Total GFLOPs** multiplies the per-epoch cost by the total epoch count reported in the main results table.

Table 7 clearly shows that CF-AMC-SSL's efficiency stems primarily from a reduced number of epochs rather than a lower per-epoch cost. All methods have the same per-view backbone cost; differences arise from (i) the number of views per image and (ii) the adversarial training strategy. PGD-5 incurs a $\sim 5\times$ compute overhead, whereas Free-AT with $m=3$ reduces this to $\sim 1.3\times$ through gradient reuse. Although multi-crop EMP variants have a higher per-epoch compute cost due to more views, CF-AMC-SSL achieves substantially lower *total* compute by converging in only 10 epochs under Free-AT (e.g., 852.8 GFLOPs per image in total) compared to PGD-based baselines or pairwise Free-AT methods that require orders-of-magnitude more total passes. This analysis quantifies both the adversarial overhead and the contribution of multi-crop design to the overall training efficiency.

### A.4 Evaluation under Additional Adversarial Attacks

We extend robustness evaluation beyond PGD by considering both gradient-based and gradient-free adversarial attacks. Table 8 reports performance under FGSM at different perturbation strengths and the Carlini–Wagner (CW) attack with 100 optimization steps, where CF-AMC-SSL consistently achieves higher adversarial accuracy than baseline methods. Table 9 presents results under the gradient-free Square Attack, confirming that CF-AMC-SSL maintains superior robustness across diverse threat models. These results demonstrate that CF-AMC-SSL is effective not only against PGD but also across a wider range of adversarial attacks.

Table 8: **Evaluation of different learning algorithms using ResNet-50 as the base encoder under additional adversarial attacks, including FGSM and CW:** This experiment provides a more comprehensive assessment of robustness beyond PGD-based evaluation, offering further evidence of the effectiveness of CF-AMC-SSL under diverse threat models.

| Models | | CIFAR-10 | | | | | CIFAR-100 | | | | |
|---|---|---|---|---|---|---|---|---|---|---|---|
| Linear Classifier | Base Encoder ResNet-50 | Clean | FGSM(4/255) | FGSM(8/255) | FGSM(16/255) | CW-100 | Clean | FGSM(4/255) | FGSM(8/255) | FGSM(16/255) | CW-100 |
| Robust Central Crop | CF-AMC-SSL (16 crops, m=7, 9 epochs) | **73.43** | **58.48** | **44.39** | **25.61** | **69.3** | **47.4** | **33.44** | **22.7** | **11.82** | **42.71** |
| | CF-AMC-SSL (16 crops, m=3, 10 epochs) | **75.89** | **60.52** | **45.84** | **27.37** | **71.99** | **53.34** | **35.16** | **23.58** | **12.75** | **46.55** |
| | SimCLR-FreeAdv (m=7, 150 epochs) | 49.49 | 39.54 | 31.01 | 18.52 | 46.12 | 25.49 | 18.43 | 13.32 | 7.43 | 22.68 |
| | SimCLR-FreeAdv (m=3, 150 epochs) | 66.89 | 49.17 | 34.63 | 18.34 | 61.31 | 38.95 | 26.18 | 16.89 | 8.06 | 34.69 |
| | VICReg-FreeAdv (m=7, 150 epochs) | 56.71 | 44.7 | 34.52 | 19.66 | 52.44 | 27.81 | 21.05 | 15.53 | 8.24 | 25.09 |
| | VICReg-FreeAdv (m=3, 150 epochs) | 71.25 | 55.14 | 42.23 | 26.64 | 66 | 45.35 | 31.64 | 21.23 | 11.14 | 40.1 |

Table 9: **Evaluation of different learning algorithms using ResNet-50 as the base encoder against gradient free Square Attack:** This experiment offers a more comprehensive evaluation of robustness beyond gradient-based attacks, providing additional evidence of the effectiveness of CF-AMC-SSL across diverse threat models.

| Models | | CIFAR-10 | | | | CIFAR-100 | | | |
|---|---|---|---|---|---|---|---|---|---|
| Linear Classifier | Base Encoder ResNet-50 | Clean | Square(4/255) | Square(8/255) | Square(16/255) | Clean | Square(4/255) | Square(8/255) | Square(16/255) |
| Robust Central Crop | CF-AMC-SSL (16 crops, m=7, 9 epochs) | **73.43** | **65.91** | **60.21** | **49.96** | **47.4** | **35.09** | **31.04** | **24.91** |
| | CF-AMC-SSL (16 crops, m=3, 10 epochs) | **75.89** | **70.15** | **63.07** | **51.53** | **53.34** | **40.27** | **35.47** | **27.49** |
| | SimCLR-FreeAdv (m=7, 150 epochs) | 49.49 | 44.88 | 42.24 | 35.04 | 25.49 | 19.6 | 17.65 | 13.84 |
| | SimCLR-FreeAdv (m=3, 150 epochs) | 66.89 | 57.44 | 51.13 | 40.47 | 38.95 | 30.85 | 27.03 | 20.99 |
| | VICReg-FreeAdv (m=7, 150 epochs) | 56.71 | 51.32 | 47.29 | 39.19 | 27.81 | 22.11 | 19.77 | 15.85 |
| | VICReg-FreeAdv (m=3, 150 epochs) | 71.25 | 65.85 | 59.51 | 48.13 | 45.35 | 34.43 | 30.49 | 24.13 |

