# OpenReview forum: "An Empirical Study of the Accuracy-Robustness Trade-off and Training Efficiency in Robust Self-Supervised Learning"
_TMLR — Accepted by TMLR_

### Review · Reviewer_XLxS · 2025-07-11

**Summary Of Contributions:**

This paper examines the integration of adversarial training with self-supervised learning, focusing on Extreme-Multi-Patch Self-Supervised Learning. The authors propose Robust-EMP-SSL, which extends EMP-SSL to adversarial training scenarios using multiple crops per image, and Cost-Free Adversarial Multi-Crop Self-Supervised Learning, which incorporates free adversarial training to reduce computational costs. Through experiments on CIFAR and ImageNet, the authors demonstrate that their methods achieve better trade-offs between clean accuracy, adversarial robustness, and training efficiency compared to robust SimCLR baselines.

**Audience:**

Yes

**Broader Impact Concerns:**

No concerns on the ethical implications of the work.

**Claims And Evidence:**

Yes

**Requested Changes:**

The requested changes are referred to the **Cons** section.

**Strengths And Weaknesses:**

**Pros**
- Practical impact: The proposed CF-AMC-SSL method effectively reduces the high computational cost of adversarial training in SSL.
- Training time is significantly reduced (e.g., from hundreds of minutes to 97) while maintaining comparable performance.
- The method is tested across multiple datasets (CIFAR-10/100, ImageNet-100) and architectures (ResNet-18/50), demonstrating some generalizability.

**Cons**
- The method primarily combines two existing paradigms, with many prior works exploring similar directions. Empirical baselines should be re-run or imported from strong, closely related methods. (1) DeACL: Decoupled Adversarial Contrastive Learning; (2) ProFeAT: Projected Feature Adversarial Training for Self-Supervised Learning of Robust Representations; (3) Enhancing Adversarial Contrastive Learning via Adversarial Invariant Regularization; (4) Adversarial Self-Supervised Contrastive Learning
- Few and outdated references (mostly pre-2023); limited discussion of foundational and efficiency-oriented SSL methods and SSL+adversarial training. The paper should clearly articulate how the proposed multi-crop + free-AT approach differs conceptually and quantitatively.
- No evaluation on modern architectures (e.g., WRN, ViT) or comparison with recent SSL frameworks under adversarial training (e.g., BYOL, SwAV, MoCo-v3, VICReg).
- While CF-AMC-SSL reports significantly reduced training time, it remains unclear how much of this gain comes from the reduced number of epochs versus the efficiency of the free adversarial training. An ablation comparing time per epoch would help clarify this. Additionally, please include computational metrics such as GFLOPs and the overhead of generating multiple adversarial examples per image.

---

> ### Author Response · Authors · 2025-08-08
>
> 1. Prior Work and Baselines
>
> Thank you for the constructive feedback. As suggested in your third comment, we have also incorporated VICReg as a point of comparison to reflect recent SSL approaches that rely solely on positive pairs, which is consistent with the direction of our work. We have also added a Related Work section following the introduction, covering recent contributions such as DeACL, ProFeAT, Adversarial Invariant Regularization, and DAQ-SDP—all of which represent valuable progress in robust self-supervised learning.
> That said, the primary objective of our study is to perform a focused comparison between EMP-SSL—which leverages multiple crops per image—and single-pair-based SSL methods like SimCLR. We aim to assess their relative strengths and weaknesses in terms of adversarial robustness and training efficiency. This focused comparison allows us to better understand the trade-offs introduced by multi-crop augmentation under Free-AT, a setting that we believe has not been systematically explored in prior SSL+AT literature.
> While the methods you mentioned are certainly impactful, incorporating them would be complementary and would expand the scope of our targeted comparison. Including them within this paper would not directly contribute to the specific research questions we seek to answer.
>
> 2. Outdated References and Contribution Clarity
>
> We appreciate this helpful observation. The revised manuscript includes updated references and a clearer articulation of our contribution. Unlike prior work that introduces new adversarial formulations (e.g., decoupling losses or perturbing feature space), we isolate the role of crop multiplicity in SSL—comparing multi-crop (EMP-SSL) to pairwise crops (SimCLR and VICReg)—under Free-AT. This clarifies how view multiplicity affects the robustness–efficiency trade-off.
> Quantitatively, our results show that multi-crop + Free-AT achieves better robustness, reduced training time, and competitive clean accuracy compared to pairwise + Free-AT. This targeted analysis complements, rather than duplicates, prior SSL+AT work.
>
> 3. Architectures and SSL Frameworks
>
> Thank you for your thoughtful comment. As you suggested, we have incorporated VICReg into our evaluation to provide a meaningful comparison with recent SSL frameworks that rely solely on positive pairs. These results are presented in the updated Table 5 and the newly added Table 8 and 9 (See appendix A.4).
> Our study is intentionally focused on isolating and analyzing the effect of multi-crop augmentations under adversarial training, rather than conducting exhaustive architectural benchmarking. We selected SimCLR as a baseline due to its conceptual simplicity and representative use of positive-negative pairs in contrastive learning.
> While modern architectures such as WRN and ViT are certainly important, adversarial training remains computationally intensive—even with standard ResNet backbones. Extending our evaluation to architectures like ViT would require substantial adjustments to both the loss design and training pipeline, which we consider a valuable direction for future work.
>
>
> 4. Efficiency Analysis and GFLOPs
>
> Thank you for raising this important point. The reported training-time reduction in CF-AMC-SSL is due to two factors:
> (1) the efficiency of Free-AT, which reuses gradients across mini-batches, and
> (2) the faster convergence enabled by multi-crop augmentations.
> As shown in Table 1, SimCLR-FreeAdv requires 167 epochs to converge, whereas CF-AMC-SSL achieves comparable or better clean and robust performance in only 10 epochs. Although per-epoch training time is higher for CF-AMC-SSL due to more views, the overall training time is much lower.
> To support this claim, we added a complexity analysis in Appendix A.3, including:
> GFLOPs per forward pass (e.g., 4.1 GFLOPs for ResNet-18),
> Adversarial overhead multipliers (e.g., PGD-5 ≈ 5×, Free-AT with m=3 ≈ 1.3×),
> Total effective FLOPs per image.
>
> We also added a min/epoch column to Table 1 to report the actual training time per epoch. This allows readers to directly assess trade-offs between per-epoch cost and convergence speed. While CF-AMC-SSL incurs a higher per-epoch cost due to multiple views, it converges in under 10 epochs—leading to significantly lower total training time than PGD-based methods or pairwise Free-AT baselines. This analysis clarifies both the adversarial overhead and the benefits of multi-crop design in achieving efficient robust learning.
>
> All changes in the revised manuscript are highlighted in orange.

---

### Review · Reviewer_H1eh · 2025-07-16

**Summary Of Contributions:**

This paper combined self-supervised learning (SSL) with adversarial training to reduce computational cost while maintaining competitive clean accuracy and robustness. To this end, the authors adopted EMP-SSL during adversarial training, which reduced the number of training epochs by increasing the number of patches per image. In addition, they incorporated free adversarial training into the multi-crop SSL framework to further cut down training time. Experiments showed that the proposed method significantly reduced the time cost of adversarial training under SSL settings.

**Audience:**

Yes

**Claims And Evidence:**

Yes

**Requested Changes:**

See Weaknesses.

**Strengths And Weaknesses:**

**Strengths:**

$\bullet$  The authors incorporate free adversarial training into self-supervised learning, significantly reducing both the number of training epochs and overall training time.

$\bullet$ The paper compares crop-based and patch-based data augmentation strategies within the EMP-SSL framework, showing that the former offers better adversarial robustness, while the latter achieves higher clean accuracy.

**Weaknesses:**

$\bullet$ In Section 2.1.2, the authors should clearly state that the objective is to **maximize** the overall loss [1], not minimize it. This ambiguity can easily confuse readers. Additionally, the dimensions of each variable (e.g., $x_j$, $x_b^i$, $Z_i$, $Z$) should be explicitly specified (e.g., vector, matrix, or tensor) to improve clarity.

$\bullet$ The presentation of results in Tables 1, 4, 5, and 6 could be reorganized. It is recommended to group baseline and proposed methods together, changing only one variable at a time for clearer comparison. For example, in Table 1, SimCLR-based methods should be grouped to show how the proposed method affects accuracy, robustness, and runtime within this category. Similarly, EMP-based methods should be grouped accordingly.

$\bullet$ The authors are encouraged to include evaluations under more adversarial attacks, such as FGSM [2] and C&W [3] to provide a more comprehensive assessment of robustness in Table 1.

[1] Tong et al. Emp-ssl: Towards self-supervised learning in one training epoch. arXiv preprint arXiv:2304.03977, 2023.

[2] Goodfellow et al. Explaining and harnessing adversarial examples. ICLR, 2015.

[3] Carlini et al. Towards evaluating the robustness of neural networks. S&P, 2017.

---

> ### Author Response · Authors · 2025-08-08
>
> 1- In Section 2.1.2, the authors should clearly state that the objective is to maximize the overall loss [1], not minimize it. This ambiguity can easily confuse readers. Additionally, the dimensions of each variable (e.g., xj, xbi, Zi, Z) should be explicitly specified (e.g., vector, matrix, or tensor) to improve clarity.
>
> Thank you for your valuable and constructive feedback. In response, we have updated Section 2.1.2 (now Section 3.1.2 in the revised version) to clearly state that the objective is to maximize the overall loss, addressing the ambiguity. We have also specified the dimensions of all variables (e.g., xj, xbi, Zi, Z) using standard notation conventions—bold lowercase letters for vectors and bold uppercase letters for matrices or tensors.
>
>
> 2- The presentation of results in Tables 1, 4, 5, and 6 could be reorganized. It is recommended to group baseline and proposed methods together, changing only one variable at a time for clearer comparison. For example, in Table 1, SimCLR-based methods should be grouped to show how the proposed method affects accuracy, robustness, and runtime within this category. Similarly, EMP-based methods should be grouped accordingly.
>
> Thank you for the helpful suggestion. Based on your recommendation, we have revised Tables 1, 4, 5, the newly added Table 8 and 9 to group baseline and proposed methods more clearly. In particular, we now organize SimCLR-based, EMP-based, and newly added VICReg-based methods into separate groups to enable more direct comparisons, varying only one factor at a time. This reorganization makes it easier to observe the impact of our proposed approach on clean accuracy, adversarial robustness, and training time within each family of methods.
>
>
> 3- The authors are encouraged to include evaluations under more adversarial attacks, such as FGSM [2] and C&W [3] to provide a more comprehensive assessment of robustness in Table 1.
>
> Thank you for your insightful comment. In response to your suggestion, we have added a new table (Table 8 and 9) that reports evaluations under additional adversarial attacks, including FGSM [2], C&W [3] and gradient-free square attack (See new added appendix A.4). This addition provides a more comprehensive assessment of robustness beyond PGD-based evaluation, offering further evidence of the effectiveness of our method under diverse threat models.
>
> All changes in the revised manuscript are highlighted in orange.

---

### Review · Reviewer_2h9S · 2025-08-11

**Summary Of Contributions:**

This paper mainly focuses on the adversarial training of self-supervised learning methods. Specifically, the authors first propose Robust-EMP-SSL, which is an adversarially trained version of the existing EMP-SSL method. With experiments, the authors show that, without adversarial training, the existing self-supervised learning methods are vulnerable to adversarial attacks. Then, the authors empirically find further improvements of Robust-EMP-SSL: data augmentation with cropped images, use of a robust linear classifier, and free adversarial training. Eventually, the authors propose CF-AMC-SSL and demonstrate the effectiveness and efficiency of the proposed method.

**Audience:**

Yes

**Broader Impact Concerns:**

I don’t see a particular broader impact concern regarding this paper.

**Claims And Evidence:**

No

**Requested Changes:**

1. I don’t think that this paper is a comprehensive empirical study. If the authors want to claim so, they should,
    * Consider more (robust) self-supervised learning methods,
    * Consider exploring additional variations of data augmentations
    * Use more attack methods
    * Use more datasets
2. In my opinion, the paper’s main achievement is the development of CF-AMC-SSL, and all the other findings are subsidiary. The authors may claim that the other details are essential because they are based on empirical studies that provide valuable insights, but I disagree. To claim an empirical study, more variations mentioned above should be covered in the experiments.
3. The proposed framework, i.e., CF-AMC-SSL with a robust linear classifier, should be compared with proper competitors.
4. Reorganize the location of tables and figures so that the readers don’t need to look back at the papers to find the tables and figures.

**Strengths And Weaknesses:**

### Strengths
1. The paper presents a detailed introduction to the methods (i.e., SimCLR and EMP-SSL), which helps readers follow the idea.
2. The paper shows various approaches to improve the robustness of self-supervised learning.

### Weaknesses
1. The experiments mainly study two methods, namely SimCLR and EMP-SSL, and their robust variations. However, experimenting on only two methods is not comprehensive enough to represent the general properties of robust self-supervised learning.
2. The authors mentioned a few research works that extended adversarial training to the self-supervised learning framework. Then, they proposed Robust-EMP-SSL to reduce the training epochs. However, I cannot find any experiments that compare the robustness/efficiency of the proposed methods to the existing robust self-supervised learning approaches.
3. This paper claims to be a comprehensive empirical study that provides different insights, but its scope is narrower than what is claimed in the paper. In my opinion, the paper’s experiments gradually converge to the development of the CF-AMC-SSL framework (with a robust linear classifier). To explain, the authors start with the idea of Robust-EMP-SSL, then each section improves the idea with crop-based data augmentations, a robust linear classifier, and free adversarial training. This paper is not about a comprehensive empirical study, but about a new method for robust self-supervised learning.
4. The performance of CF-AMC-SSL should be compared with more existing methods for robust self-supervised learning. The paper compares its performance only with that of some other encoders with free adversarial training. However, other robust self-supervised learning methods should be compared to the proposed method.
5. The paper's contents are not well-organized, and readers should read the paper back and forth to locate the tables or figures that the paper refers to. This makes the paper harder to read.
6. Most experiments use only two datasets: CIFAR-10 and CIFAR-100. Because both datasets consist of the same input data (with different labeling), the input distribution is essentially the same in those experiments. Only Table 4 contains (partial) evaluation on ImageNet-100. However, there are too many missing evaluations, and this makes the experiments less consistent.
7. Even after the revision, the paper did not evaluate its performance against various attack methods. First, PGD generalizes FGSM, so adding FGSM does not provide any insights beyond the PGD-based evaluations. Including CW (in addition to PGD and AA) could be sufficient for usual papers that propose robust learning methods, but I expect more variations, e.g., gradient-free attacks, for this long paper.

---

> ### Author Response · Authors · 2025-08-21
>
> We thank the reviewer for their thoughtful and detailed feedback.
>
> 1. On whether this is a “comprehensive empirical study” or a method paper:
>
> Our intent in framing the paper as an empirical study was to emphasize that CF-AMC-SSL emerged through a sequence of controlled experiments addressing targeted research questions in adversarially robust self-supervised learning (SSL). Specifically, we investigated:
>  - Augmentations: crop- vs. patch-based augmentations and the impact of varying the number of crops/patches.
>  - Efficiency: training epochs and the integration of Free Adversarial Training (FAT).
>  - Evaluation: central-crop vs. multi-crop embedding aggregation and the role of robust linear evaluation (r-LE).
>  - Generalization: comparisons across datasets (CIFAR-10, CIFAR-100, ImageNet-100), architectures (ResNet-18, ResNet-50), and multiple attacks (PGD, AutoAttack, CW).
>
> While these experiments converge on CF-AMC-SSL as the most effective framework, the ablations provide broader insights into:
>  - Accuracy–robustness–efficiency trade-offs in adversarial SSL.
>  - The effect of augmentation count on robustness.
>  - The influence of evaluation protocols (central crop, multi-crop, r-LE).
>  - The role of FAT in robust SSL.
>
> We acknowledge that if “comprehensive” is interpreted as covering all robust SSL methods and datasets, our scope is bounded. To avoid misunderstanding, we can revise the title and introduction to clarify that the paper is both an empirical investigation and a method proposal, and that “comprehensive” refers to systematic coverage within our design space rather than exhaustiveness across the entire field.
>
> 2. Reorganization of tables and figures:
>
> We appreciate this suggestion and have reorganized the placement of all tables and figures for better readability in the revised version.
>
> 3. Attack methods:
>
> We evaluated robustness against PGD, CW, and AutoAttack, which are among the most widely adopted adversarial attacks in the robust learning literature. Following your suggestion, we also included evaluations with the gradient-free Square Attack. Additional results against FGSM, CW, and Square attacks are provided in Appendix A.4.
>
> 4. Use of datasets:
>
> We appreciate the observation regarding CIFAR-10 and CIFAR-100 sharing input distributions. While this is true, we note that the larger number of classes in CIFAR-100 can make adversarial evaluation more challenging, as the adversary has more opportunities to alter predictions. Because adversarial training requires long training times, it is common practice to focus on CIFAR-10 and CIFAR-100. Still, to better demonstrate the generalization of our claims, we included experiments on ImageNet-100. Following your suggestion, and within the available timeline, we were able to complete the additional experiments reported in Table 4.
>
> 5. Comparisons with other SSL methods:
>
> In the original EMP-SSL paper, comparisons were made with SimCLR, VICReg, and BYOL. In our work, we maintained methodological consistency by integrating FAT into SimCLR, VICReg, and EMP-SSL, ensuring that improvements are attributable to our proposed crop-based adversarial extensions rather than unrelated design choices.
> Other robust SSL frameworks, such as Adversarial Contrastive Learning (ACL) [1] and Adversarial Invariant Regularization (AIR) [2], are largely orthogonal. For example:
>  - ACL modifies the loss function by mixing natural–natural and natural–adversarial pairs.
>  - AIR adds modular style-invariance penalties.
> Both can be layered on top of EMP-SSL, SimCLR, or VICReg. Thus, a direct comparison would not isolate the specific contributions of multi-crop augmentation and FAT, which are the focus of this work.
>
> 6. Data augmentation variations:
>
> We agree that adversarial SSL could, in principle, be studied with more augmentation families (e.g., color jitter, blur, solarization). However, such exhaustive exploration is computationally prohibitive in adversarial training. We therefore focused on crop vs. patch augmentations and number of crops, motivated by Bag of Image Patches (BoP) work [3] that identified these as particularly impactful in SSL. Our extension shows that, under adversarial training, crop-based augmentations consistently outperform patch-based ones, an insight directly relevant to robust SSL design.
>
>
> [1] Jiang, Ziyu, Tianlong Chen, Ting Chen, and Zhangyang Wang. "Robust pre-training by adversarial contrastive learning."
>
> [2] Xu, Xilie, Jingfeng Zhang, Feng Liu, Masashi Sugiyama, and Mohan S. Kankanhalli. "Enhancing adversarial contrastive learning via adversarial invariant regularization."
>
> [3] Chen, Y., Bardes, A., Li, Z. and LeCun, Y., 2022. Bag of image patch embedding behind the success of self-supervised learning.
>
> All changes in the revised manuscript are highlighted in orange.

---

### Decision · Action_Editor_w8x8 · 2025-09-30

**Recommendation:** Accept with minor revision

**Additional Comments:**

The work is correct and informative, with agreement of the claims and evidence, and with an audience of researchers interested in self-supervised learning, adversarial optimization, and robustness. The authors have already engaged with the reviewers and their feedback to improve the work and so acceptance with minor revision is recommended for the incorporation of any last changes. In particular the action editor suggests potential revisions to re-phrase and re-organize as detailed by 2h9S and H1eh, which is not so much a change in content, but in presentation. The points raised by XLxS have already been incorporated.

The action editor thanks the authors and reviewers for their full participation in the TMLR process and congratulates the authors on acceptance of the paper.

**Audience:**

Yes

**Audience Explanation:**

The separate topics of self-supervised learning and adversarial training are established focuses in current research on machine learning. Their intersection is already examined by existing work in TMLR, ICLR, NeurIPS, and ICML and so an audience on these topics from these venues is expected. More specifically, this work extends and further studies an established method (EMP-SSL), and so can be of interest to those working on related methods and existing methods of this type.

All reviewers agreed on the satisfaction of the audience criterion.

**Claims And Evidence:**

Yes

**Claims Explanation:**

This work contributes to the intersection of self-supervised learning and adversarial optimization to contribute a robust extension of multi-patch self-supervised learning and new empirical information comparing multiple robust and non-robust self-supervised learning methods. The claims are scoped to the multi-patch combination of multi-patch methods with adversarial training to inform the community on the impacts on efficiency, accuracy, and robustness (= accuracy under adversarial attack). The main results are that the proposed Robust-EMP-SSL and CF-AMC-SSL better balances accuracy and robustness while converging in fewer steps of optimization. The evidence is provided in measurements of accuracy on standard datasets CIFAR-10/100), with standard attacks (PGD + and the fuller AutoAttack suite), and metrics of accuracy and computation time (minutes and epochs).

All reviewers agreed on the satisfaction of the claims and evidence criterion.